# Structure of ABCB1/P-Glycoprotein in the Presence of the CFTR Potentiator Ivacaftor

**DOI:** 10.3390/membranes11120923

**Published:** 2021-11-25

**Authors:** Alessandro Barbieri, Nopnithi Thonghin, Talha Shafi, Stephen M. Prince, Richard F. Collins, Robert C. Ford

**Affiliations:** 1School of Biological Sciences, Faculty of Biology Medicine and Health, The University of Manchester, Oxford Road, Manchester M13 9PT, UK; alessandro.barbieri@postgrad.manchester.ac.uk (A.B.); nopnithi@g.swu.ac.th (N.T.); talha.shafi@manchester.ac.uk (T.S.); steve.prince@manchester.ac.uk (S.M.P.); Richard.Collins@manchester.ac.uk (R.F.C.); 2Bioinformatics Institute (BII), Agency for Science, Technology, and Research (A*STAR), 30 Biopolis Street, #07-01 Matrix, Singapore 138671, Singapore; 3Department of Biology, Faculty of Science, Srinakharinwirot University, 114 Sukhumvit 23, Wattana District, Bangkok 10110, Thailand

**Keywords:** P-glycoprotein, ABCB1, ABCC7, ABC transporter, ivacaftor, drug binding, Volta phase plate

## Abstract

ABCB1/P-glycoprotein is an ATP binding cassette transporter that is involved in the clearance of xenobiotics, and it affects the disposition of many drugs in the body. Conformational flexibility of the protein within the membrane is an intrinsic part of its mechanism of action, but this has made structural studies challenging. Here, we have studied different conformations of P-glycoprotein simultaneously in the presence of ivacaftor, a known competitive inhibitor. In order to conduct this, we used high contrast cryo-electron microscopy imaging with a Volta phase plate. We associate the presence of ivacaftor with the appearance of an additional density in one of the conformational states detected. The additional density is in the central aqueous cavity and is associated with a wider separation of the two halves of the transporter in the inward-facing state. Conformational changes to the nucleotide-binding domains are also observed and may help to explain the stimulation of ATPase activity that occurs when transported substrate is bound in many ATP binding cassette transporters.

## 1. Introduction

P-glycoprotein (P-gp) is an ATP-dependent multi-drug transporter of the ATP-binding cassette (ABC) family that is involved in the active efflux of many drugs and xenobiotics from cells [1]. Overexpression or activation of P-gp can lead to multi-drug resistance in cancer cells [2,3,4,5]. Drugs and inhibitors are thought to bind to the inward-facing conformation of P-gp [6,7,8,9], which is more favored in the post hydrolytic or nucleotide-free conditions [10,11,12]. The drug is then translocated across the plasma membrane by transition of the protein to the outward-facing state, which is more favored when ATP is bound [12]. The dynamic conformational nature of P-gp (and other ABC family members [13]) has challenged structural studies. Nevertheless, P-gp structures have been obtained by X-ray crystallography [14,15,16,17,18] and cryo-electron microscopy (cryo-EM) [7,8,9,19,20,21]. Stabilizing antibodies and ATP hydrolysis-inactivating mutations have been exploited to generate ~4 Å resolution structural data by cryo-EM [7,9,19]. Similarly, trapping the protein in a post-hydrolytic state with vanadate and ATP yielded an ~8 Å resolution structure [21]. However, under these experimental conditions, allosteric changes could be altered or inhibited.

Recent studies have demonstrated that the novel cystic fibrosis (CF) therapeutic ivacaftor is a competitive inhibitor or transported substrate of P-gp [22,23]. This drug is a potentiator of the channel activity of an P-gp homolog, the cystic fibrosis transmembrane conductance regulator CFTR/ABCC7 [24], and ivacaftor is now widely used in the clinic, both on its own as well as in combination with CFTR-corrector compounds such that roughly 90% of CF patients may now be treatable [25,26]. A prior structure for ivacaftor bound to CFTR showed the binding mode of the drug, which was on the external surface of the membrane-spanning region and within the hydrophobic region of the detergent micelle [27]. The drug itself is highly hydrophobic and was used at a relatively high concentration, but mutagenesis studies and experiments with another potentiator drug with a slightly different structure were undertaken to provide further evidence that the position of the drug in CFTR represented a common binding mode. A further interesting observation in the study was that there was minimal conformational change upon drug binding, and because of the rigid nature of the drug, conformational selection (of the drug) may not occur. Structure–activity relationships (SAR) of ivacaftor homologs with potentiating and inhibiting properties identified the quinoline nitrogen as crucial for activity via hydrogen bonding. π-π stacking with nearby residues by the same quinoline group was also found to be important. The planar nature of the drug was also shown to be a significant factor in the SAR study. The hydroxyl group of the hydroxyphenyl moiety is polar with a computed pK_a_ of 11 (CHEMBL2010601) and points towards the guanidino group of R933 in the structural model [24,27], implying polar interactions that are too distant (6 Å) for hydrogen bonding.

In this study, we examined the P-gp structure in the presence of ivacaftor. In order to minimize non-specific binding of the hydrophobic drug, we employed a concentration likely to be encountered in vivo [28] and at a sub-stoichiometric ratio with the protein. This approach had the advantage of allowing both drug-bound and drug-free forms of the protein to coexist as a mixture in the protein/drug solution immediately prior to flash freezing and cryo-electron microscopy. We reasoned that if drug binding caused a conformational change in P-gp, we should be able to distinguish different conformations using 3D classification of the particles. If no major change in the overall 3D structure occurred (as was the case for CFTR [27,29]), we expected to obtain a single 3D structure but with fractional occupancy of the drug at its binding site, leading to weak density for ivacaftor in the experimental 3D density map. A third possibility existed: that drug binding would cause conformational changes, but that discrimination of these different conformations would not be possible by the image processing routines. In this case, we would expect a smeared, low-resolution 3D map for the protein with no information about drug binding. In order to minimize the chances of the latter scenario, we employed the Volta phase plate to collect very high signal:noise data with minimal under focus. There are difficulties in the use of the Volta phase plate that restrict its current wide application [29], and there are currently only a few examples of studies resulting in resolution better than 3 Å [29]. Nevertheless, previous studies with 3D maps of resolution similar to those reported here have allowed drugs and small ligands such as sterols and nucleotides to be positioned and their binding interactions modeled within the protein of interest [30,31,32,33,34,35,36,37,38,39,40,41,42]. In this report, we aim to compare apo and drug-bound states within a single cryo-EM specimen. If successful, the approach we describe should minimize specimen preparation and biological variation as well as allowing relatively physiological levels of drugs to be employed.

## 2. Materials and Methods

*P. pastoris* harboring the *opti-mdr3* gene was kindly provided by Prof. Ina L. Urbatsch, Texas Tech University [43]. Cell culture was performed using the shake-flask method described by Beaudet and co-workers with minor modifications [44]. Overexpression of the gene product (*abcb1a*) was initiated by the addition of methanol, and the expression was prolonged following an optimized procedure [21]. Cell pellets were collected and stored at −80 °C.

Cell rupture was conducted using glass bead beating, and microsomal membranes were prepared as described in (4). Microsomal membranes were diluted to 2.5 mg/mL prior to solubilization in detergent-containing buffer (50 mM Tris pH 8.0, 10% (*v*/*v*) glycerol, 50 mM NaCl, 1 mM 2-mercaptoethanol and 2% (*w*/*v*) DDM). Solubilized materials were processed through multi-step protein purification, described previously [21,22]. Purified protein was concentrated to 5–10 mg/mL using a Vivaspin concentrator with 100-kDa cut-off, flash-frozen in liquid nitrogen, and stored at −80 °C.

Drug binding and P-gp thermostability were assayed in 3 ways using an UNCLE (Unchained laboratories) instrument: concentrated P-gp protein (1 µg per sample) was diluted into 10 µL of buffer (100 mM Tris-HCl pH 8, 150 mM NaCl, 10% glycerol, 0.1% DDM and 0.02% CHS) containing different concentrations of ivacaftor. The experiments were also repeated with the further addition of 100 ng of CPM dye (7-Diethylamino-3-(4′-Maleimidylphenyl)-4-Methylcoumarin), which acts as a reporter of solvent-exposed cysteine residues [22,45,46,47]. The mixtures were loaded into capillaries at 4 °C, and the latter were inserted into pre-chilled 16 well UNCLE capillary cassettes. Fluorescence emission spectra were recorded between 200 nm and 700 nm with an excitation wavelength of 266 nm. Static light scattering was recorded simultaneously using the 266 nm laser source. Data were gathered between 16 °C and 90 °C with a stepwise increase in the temperature (2 °C increments) followed by the recording of spectra for all the samples. The total heating run required approximately 70 min. Raw data were exported and then analyzed for tryptophan fluorescence, static light scattering, and CPM fluorescence using the GraphPad Prism software package.

P-gp was initially buffer-exchanged into a detergent/glycerol-free buffer. The protein was diluted to 1.1 mg/mL and incubated with 2 µM Ivacaftor (stock: 100 µM in DMSO) for 30 min on ice. Quantifoil 200 or 400 Au grids with a 1.3-micron spacing pattern were pre-treated prior to protein deposition, as described previously [21]. Briefly, grids were surface-cleansed by multiple chloroform washes to eliminate hydrophobic residues, then glow-discharged. An FEI Vitrobot MkIV was employed to facilitate sample vitrification where 3 µL of protein was gently loaded onto the middle of the grid, immediately blotted for 6 s, and flash-frozen in liquid ethane. Grids were assessed for specimen quality (i.e., ice thickness, particle distribution, etc.) using a Polara G30 before shipping to the eBIC UK National facility for high-resolution data acquisition on an FEI Titan Krios G2 microscope. Data were recorded at 300 kV using a 20 mV energy filter and a Gatan K2 electron detector. Images were acquired via the Volta phase plate [48] with a phase shift range between 0.15 and 0.8 radian and fixed defocus at −0.5 µm. A total of ~64 e^−^ Å^−2^ were used over 40 frames spanning 10 s of exposure, and early and late frames were discarded. The magnification was calibrated at 1.043 Å/pixel. A total of 2436 movie stacks were collected, and beam-induced image shift was eliminated using MotionCorr2 [49].

The data processing of the 2436 images was entirely performed with *cis*TEM [50]. A total of 1030 images were eliminated for possessing poor CTF fitting profiles and exceeding defocus values. The final set of 1406 images were divided into 2 similarly-populated subsets (696i and 710i) to independently assess the reproducibility of the 2D classification procedure. Ab-initio 3D reconstruction was performed employing ~50,000 particles derived from several 2D classification runs of the 696i subsets. For each of the 5 classes that were generated ab-initio, a PDB model was built using Molecular Dynamics Flexible Fitting (MDFF) with NAMD 2.12, as detailed previously [21]. These models were used as a measure for comparing NBD separation in the five classes. Ab-initio 3D maps were then low pass filtered to 10 Å and employed as starting templates for 3D refinement using the full final set of 104,000 selected particles. After full 3D refinement of the maps, real-space refinement of atomic models against the maps was performed within the PHENIX package [51], with the MDFF-derived models used as the initial inputs. Half-maps and Fourier shell correlation curves were generated employing the *cis*TEM and *calculate_fsc* routine. Local resolution was also estimated using the program ResMap [52]. Rigid-docking of ivacaftor was performed with the fit-in-map routine of UCSF Chimera [53] with a 5 Å-resolution simulated map of ivacaftor atoms at 0.252 (7 × SD) and with optimization for overlap with the experimental (additional) density in the P-gp central cavity. Fitting was subsequently minimized in Schrodinger’s Maestro (Schrödinger Release 2021-2: Maestro, Schrödinger, LLC, New York, NY, USA, 2021), concomitantly allowing the rotamer adjustments of neighboring residues. The final result was verified to maintain the occupancy of the cryo-EM density. Nucleotide-binding pocket volumes of NBD1 and NBD2 were estimated by summation of all voxels with lower than mean density values that were within 5 Å of a positioned nucleotide. This was carried out with the *matchmaker, color zone/split map*, and *calculate volume and area* functions of Chimera [52] and using the NBDs in the 6q81 P-gp atomic model for positioning [21].

## 3. Results

### 3.1. Ivacaftor Binding Affinity of Murine P-gp

Ivacaftor has been identified as a competitive inhibitor of Hoechst 33,342 and digoxin transport by human P-gp [22,23]. Ivacaftor stimulated human P-gp ATPase activity eight-fold, suggesting it could also be considered as a transported allocrite, and its dissociation constant (K_D_) was estimated at 0.3 μM using a fluorescent dye transport assay with the purified protein [22]. Similarly, a half-maximal inhibitory concentration (IC_50_) of 0.2 μM was estimated using an in-vivo assay [23]. We tested ivacaftor binding to murine P-gp using various methods: Static light scattering and tryptophan fluorescence changes yielded K_D_ values of 0.2 μM (static light scattering) and ≅1 μM (tryptophan fluorescence) (Appendix A). The binding of a sulfhydryl-reactive fluorescent dye (7-Diethylamino- 3-(4’-Maleimidylphenyl)-4-Methyl-coumarin, CPM) to solvent-exposed P-gp cysteine residues detected an unfolding transition between 30–50 °C that was sensitive to ivacaftor with a K_D_ estimated at 0.2 μM (Appendix A). Hence, we concluded that the affinity for ivacaftor of murine P-gp was similar to human P-gp. Moreover, the detection of light scattering changes upon ivacaftor binding implied conformational changes in P-gp (e.g., due to an induced fit). Cryo-EM studies were subsequently performed with a concentration of ivacaftor (2 μM) that was well above the K_D_, although at this level it was still at a sub-stoichiometric ratio versus protein (estimated at 1.1 mg/mL, or roughly 8 μM using a molecular mass of 150kDa for P-gp).

### 3.2. Cryo-EM of P-gp in the Presence of Ivacaftor

Images (2436) were recorded with the Volta phase plate, and after initial rejection of poor quality images by eye, the contrast transfer function (CTF) of each was estimated. A total of 1537 images gave CTF fits that were judged to be reliable whilst 868 were rejected at this stage. A second criterion was applied to eliminate all images affected by excessive defocus values, and a total of 131 further images were eliminated, with a final set of 1406 images (Appendix A). As expected, the P-gp particles were readily identifiable in the images in the inward-facing conformation (Appendix A), and automated particle picking was performed. Reference-free classification of the projections of the resulting picked particles allowed a further pruning of the dataset by deleting any non-P-gp contaminants as well as small P-gp aggregates (Appendix A). Several iterations of 2D classification were completed to remove bad particles, and this was judged to be complete when consistent 2D classes were obtained. The reproducibility of the 2D classification was then assessed by splitting the dataset into two roughly equally populated sub-sets, which individually showed the same outcome (Appendix A).

A*b-initio* 3D classes of the final P-gp dataset were generated from a subset of only ~50,000 particles, and these are shown in Appendix A. At this early stage, the 3D classes were relatively low resolution (~7 to 8 Å, as estimated by Resmap [54]). Nevertheless, the two procedures clearly identified only inward-facing conformations, even when five classes were allowed. Sets of particles displaying a wider separation of the NBDs (classes *a* and *c*) were also picked out by the image processing package. For each of the five low-resolution ab-initio maps, a PDB model was derived using Molecular Dynamics Flexible Fitting (see Methods), and results were compared. Utilizing the center of mass of the NBDs as a reference, classes *b*, *d,* and *e* showed a narrower separation of the NBDs (from 56 Å to 53 Å), while both classes *a* and *c* were found to have a wider distance (60 Å). A moderate low-pass filter was applied to the *ab-initio* maps to avoid over-fitting effects. CTF parameters were also further refined at this stage with per-micrograph CTF fits using the dedicated *cis*TEM functionality for this. Refinement of the full dataset (104,000 particles) starting with each of the five *cis*TEM *ab-initio* 3D classes was carried out with a combination of auto- and manual-refinement, and an atomic model was generated for each resulting refined 3D map using the PHENIX *real space refinement* routine [51]. The 3D maps, compared at a density threshold yielding a map volume of 153,000 Å^3^ are displayed in Figure 1.

The resolution of each map was assessed by splitting each data set in half and separately refining each half dataset. The correspondence between half-maps was analyzed in reciprocal space using Fourier shell correlation (FSC) as well as using the Resmap routine, which calculates local resolution in the maps [54,55]. The resolution assessments as well as other data pertaining to the five refined maps and the five atomic models, are summarised in Table 1. Each 3D map could be generated from fewer particles than in prior studies of P-gp by cryo-EM [7,19,21] due to the high contrast of individual particle projections produced with the Volta phase plate. However, the resolution achieved was estimated at 4–6 Å for all the maps (Table 1, Appendix A). Map resolution was not correlated with particle numbers, consistent with the idea that, for structural studies of unstabilized P-gp, intrinsic protein flexibility is likely to be a major factor in determining the resolution achievable. Interestingly, map *a* displayed the lowest resolution of all the maps in the various tests employed, although the overall map-to-model correlation was better (Table 1, Appendix A). The Resmap analysis of map *a*, implied that variation between the half-maps was mainly confined to the periphery of the NBDs and the micelle (Appendix A). Conversely, map *e*, with the closest approach of the NBDs, was estimated to have the highest resolution by the Resmap algorithm [54] and by Fourier shell correlation [55] (Appendix A). In general, the map-to-model correlation was lower for the NBDs compared to the TMDs in all maps (Appendix A).

### 3.3. Interpretation of the Differences between the 3D Maps

The various 3D maps showed the typical hinging of the two halves of the molecule, giving differences in NBD separation, but it was apparent that NBD1 flexing relative to the other P-gp domains was also being differentiated by the 3D classification algorithm: NBD1 appeared to progressively rotate upwards towards the TMDs when one examines maps *a* through *e*; (Figure 1, upper panel). Separation of NBD1 from NBD2 was greatest for maps *a* and *c*; was narrower in maps *b* and *d*; whilst map *e* had the closest approach of the NBDs (Figure 1, upper panel). As previously reported [21], densities assigned to the N-terminus and to the end of the NBD1-NBD2 linker on the opposite side of the molecule can be observed in the maps, especially map *a* (Figure 1 upper panel). Maps *b* and *d* showed additional density for the linker but lacked density for the N-terminal region. In contrast, map *c* showed the latter but appeared to lack the linker density. The density for the extracellular loop (ECL) between TM helices 1 and 2 was weaker for maps *c* and *e*, perhaps representing increased disorder in this loop, which may be core glycosylated in the expression system used. These symmetry-breaking features of P-gp likely aid the image processing via discrimination of projections of alternative 180°-rotated orientations of particles (Appendix A).

The transmembrane regions (Figure 1, lower panel) also showed some subtle differentiation between the five 3D maps. The diameter of the micelles in maps *a* and *c* are noticeably larger; whilst map *e* has the smallest micelle, displaying some distortion from a circular annulus. These differences may be a reflection of the degree of separation of the two halves of the protein, which is greatest for maps *a* and *c* and least for map *e*. The latter map shows the most compaction of the transmembrane α-helices. The transmembrane regions of all five maps (Figure 1, lower panel) displayed the large central aqueous cavity that connects to the cytoplasm as well as to the lipid bilayer (via the two lateral gaps formed between TM helices 4 and 6 and between 10 and 12). These lateral gaps are a common feature of type IV ABC transporters such as Sav1866, MsbA, and P-gp [56]. In P-gp the central aqueous cavity has been found to house the binding sites for transported drugs and inhibitors [7,14,16]. We, therefore, closely examined this cavity for evidence of ivacaftor binding in the five maps. The criteria we used were that: (i) any features due to ivacaftor should be visible at the same density level used for the protein (ii) that at this density level, the feature(s) should be roughly rod-shaped, and (iii) have a continuous volume equivalent in size to that expected for the drug. Finally, (iv) once fitted, the ivacaftor molecule should not have serious clashes with nearby residues in a fitted P-gp atomic model (i.e., none that could not be remedied with rotamer adjustment). Only map *a* contained a single feature that satisfied all these criteria (blue mesh, arrow, Figure 1, lower panel). All the other maps did show small features in the aqueous central cavity (Figure 1 lower panel), especially at lower density thresholds, but none of these met the first three criteria.

The additional density in the aqueous cavity of map *a* has a rocket-shape with two fin-like protrusions at one end. At the opposite end to the protrusions, the density is close to TM6, and in the fitted atomic model, it was close to F339 and Q343 (Figure 2). In accordance with the last criterion, a single ivacaftor molecule could be fitted into this density, and the overall rocket shape favored a fit with the 2,4 *tert*-butyl groups of the ivacaftor molecule corresponding to the two ‘fins’ of the rocket-shaped density (CC = 0.960 vs. CC = 0.938 for the 180° rotated version). The density is surrounded by residues contributed by TM helices 6, 10, and 12, and hence is positioned more in one lobe of the aqueous cavity (Figure 1). Residues F339 and Q343 within TM6 in the refined atomic model are close enough to form π-π (F339) and H-bonding (Q343) interactions with the quinoline group of the modeled ivacaftor molecule (Figure 2), and the phenolic hydroxyl group of the drug could also form H-bonding interactions with E871 within TM10 in this model. Small residues G868 and G985 in TM helices 10 and 12, respectively, allowed the fitting of the bulky *tert*-butyl groups of ivacaftor without steric clashes with the protein. Although the cavity is aqueous, I864, M872, L875 in TM10, and A981 in TM12 form a hydrophobic patch on its surface that appears to be compatible with the positioning of the hydrophobic drug (Figure 2). K185 on the cytoplasmic side of TM3 in the aqueous cavity is within ~10 Å of the additional density. This is somewhat reminiscent of R933 in human CFTR [27].

### 3.4. Conformational Changes Associated with the Additional Density

Assuming the additional density in map *a* is due to drug, then its presence is associated with a wider separation of the two halves of P-gp, similar to observations for the co-crystal structures of murine P-glycoprotein with the marine pollutant and P-gp inhibitor BDE-100 [15,16,57]. Similarly, widening of the structure of the TmrAB heterodimeric peptide exporter was associated with substrate-binding [13]. We, therefore, examined whether other local changes in individual domains could be identified as being specific to map *a* versus the other four maps. In this line of conjecture, we assumed that maps *b* to *e* were representative of multiple conformations of the *apo*-state. When map *a’s* atomic model was compared to the one most closely matching it: map *d* model (rmsd =1.17 Å for 1018 atom pairs within 2 Å, 1.46 Å overall 1182 atom pairs), a local bulging outwards of the transmembrane helices were apparent (see Figure 3), similar to that observed for TmrAB [13]. When we compared map *a’s* model to that of map *e* (which was the most different—rmsd 1.40 Å over 519 atom pairs within 2 Å, 2.64 Å overall 1182 atom pairs), the outwards bulging of the TM helices upon ivacaftor accommodation was more noticeable. In both cases, TM helices 1, 2, 7, and 8 move most whilst helices 6 and 12 move little (Figure 3b,c), and the bulge of the TMDs resulted in a wider separation of the NBDs. As previously mentioned, NBD1 also showed a downward rotation away from the TMDs (Figure 3).

The net result of this movement of the NBDs in the atomic models was a downward (cytoplasmic) shift of the Walker A and B regions relative to intracellular loops 1 and 4 that connect NBD1 from the TMD, and in map *a* this is manifested as an opening up of the nucleotide-binding site in NBD1 (Figure 3d). Similar changes but with a smaller magnitude, were also observed for NBD2 in the maps and atomic models. Caution must be exercised when extrapolating from atomic models that have been fitted to worse-than-atomic resolution maps, but if these conformational shifts in the NBDs are valid, then they may be reflective of allosteric changes that lead to increased ATPase activity of P-gp upon binding of a transport substrate.

## 4. Discussion

Structural data for 3D conformations of P-gp were obtained with the Volta phase plate; and with this methodology the resolution of the 3D information obtained was around 6-4 Ångstrom. Whether these resolution limitations can be overcome with a larger dataset remains to be established. The use of sub-stoichiometric levels of ligand may be a suitable approach in future cryo-EM studies, provided that ligand–protein complexes can be distinguished from ligand-free protein on the basis of different conformational status and/or by a sufficiently large additional density due to the ligand. Clearly, atomic details of ligand binding mode will depend on close to atomic resolution data, but the overall location of the ligand and some modeling of likely binding modes will be possible at intermediate resolution, as described here and in other studies for various other proteins and ligands [30,31,32,33,34,35,36,37,38,39,40,41,42].

Drug and inhibitor binding locations in P-gp have previously been studied for both mouse and human versions of the protein [8,9,15,16,58] as well as for a human/mouse chimeric construct [7]. For mouse P-gp, published structural data from X-ray crystallography is available for cyclic peptide inhibitors and for a marine pollutant BDE-100 that is also thought to act as an inhibitor of the protein [15,16]. For human P-gp, cryo-EM data are available for well-characterized substrates such as taxol [8,9] and vincristine [8] and also for some inhibitors such as elacridar, zosuquidar, and tariquidar [8]. Ivacaftor has been shown in-vitro to stimulate P-gp ATPase activity and to reduce transport by P-gp of Hoechst 33342 [22]. In humans, ivacaftor administration caused increased blood digoxin levels [23], implying inhibition of P-gp. These studies suggest that ivacaftor can compete with Hoechst 33342 transport and digoxin transport by P-gp, hence ivacaftor could be transported by P-gp. Alternatively, the data could be consistent with ivacaftor acting as a non-competitive or uncompetitive inhibitor of the protein’s transport (but not ATPase) function. The Hoechst 33342 binding site (H-site) has been modeled to be asymmetrically–located in P-gp [59,60], but this position is in the opposite lobe of the internal cavity compared to the modeled site for digoxin (D-site) [60], and the density associated with ivacaftor described here. Hence from the data and models, one could propose that ivacaftor may compete for both digoxin binding and transport. Whereas for Hoechst 33342, transport, but not binding, would be affected by ivacaftor presence. These suggestions remain to be tested experimentally.

A comparison of murine P-gp models both with and without bound compounds is shown in Figure 4, panels a–c. Their conformations are similar to the murine P-gp model for map *a* that is described here, with root mean squared deviation (rmsd) between the main chain Cα atoms of the models of around 4 Å over 1179 comparisons (i.e., not including disordered linker and N- and C-terminii atoms). When these various atomic models are superimposed (Figure 4, upper panels), the positions of the inhibitors and drug (ivacaftor) are all found within the central aqueous cavity. However, the BDE-100 and cyclic peptide inhibitors appear to be located further towards the extracellular side of the membrane.

A comparison of the mouse/ivacaftor P-gp model with human and human/mouse chimera P-gp models in the presence and absence of other small molecules is shown in Figure 4, panels d–f. Separate alignment of the two ‘wings’ of the inward-facing conformation of the mouse/ivacaftor structure was required for this comparison because the human and chimera P-gp models all show a closer approach of the NBDs. In these comparisons, both substrates and inhibitors co-localize to the same approximate position within the central cavity. As discussed by Nosol et al. [8], whilst the P-gp inhibitors display two separate molecules in the cavity, the transported substrates (and ivacaftor) display only one molecule in the cavity. Density associated with ivacaftor in map *a* is consistent with a single molecule, but again, in this comparison, it would be located closer to the cytoplasmic side of the membrane than the other compounds, with the exception of one of the elacridar molecules (orange stick representation, Figure 4).

Taken together, the data for ivacaftor is equivocal: the single density observed would imply that ivacaftor shares P-gp substrate characteristics, whilst its overall location in one lobe and over towards the cytoplasmic leaflet of the bilayer is suggestive of binding characteristics of an inhibitor according to the work of Nosol et al. [8]. It seems possible that the low concentration/stoichiometry of the drug employed in this study may result in a single bound ivacaftor molecule at a higher affinity site, even if it were an inhibitor with two potential sites. However, it should be noted that not all inhibitors appear to bind in the same position: The cyclic peptide inhibitors and BDE-100 studied with murine P-gp occupied a location different from the elacridar, tariquidar, and zosuquidar inhibitors [14,15]. In addition, of note is that in the human P-gp models, transmembrane helices 4 and 10, were kinked inwards and appear to partially close off the internal cavity on the cytoplasmic side. This behavior has not been observed thus far in the murine P-gp models (Figure 4). The closest approach of the two halves of the human and chimera P-gp atomic models is also noteworthy, although this may result from different solubilization methods employed (lipids in the nanodiscs for human P-gp; detergent in micelles for murine and chimera P-gp). Whether these differences between human and murine P-gp structures represent interspecies variation, or instead alternative conformational states in the transport cycle remains to be established. At first sight, it would seem somewhat counterintuitive that drug binding to P-gp would induce a further separation of NBDs, rather than bringing them towards the NBD dimerized and outward-facing state. However, other studies have also detected similar conformational shifts (e.g., [13,15,16], including P-gp structures co-crystallized with BDE-100. We would propose that the opening of the NBD1 nucleotide-binding site may be a precursor to further conformational shifts induced by nucleotide that then promote the formation of the outward-facing state.

## Figures and Tables

**Figure 1 membranes-11-00923-f001:**
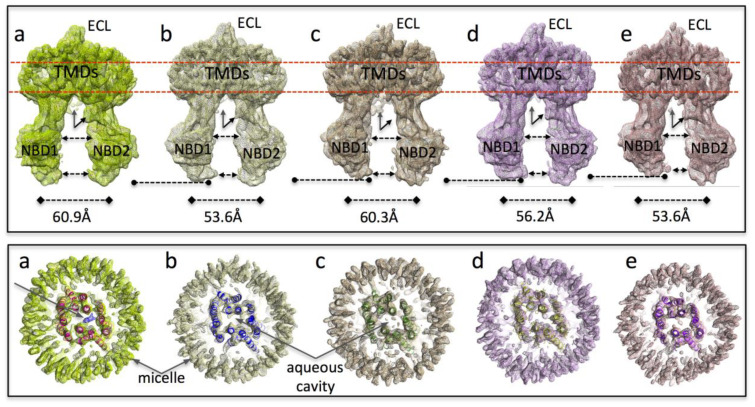
Differentiation of the five 3D maps (**a**–**e**): Maps are displayed at a density level enclosing a volume of 153,000 Å^3^. Upper panel: (1) NBD1 rotates and hinges upwards towards the TMDs; with this increasing from maps (**a**) through to (**e**), (dashed line with circles indicates the lower edge of NBD1). (2) NBD1-NBD2 separation, (dashed lines with arrowheads) is greatest for map (**a**,**c**), least for map (**e**) and intermediate for maps (**b**,**d**). (3) Density for the N-terminus (rearwards, vertical arrow) and the end of the NBD1-NBD2 linker (slanted arrow, front) is present in map (**a**), whilst other maps lack the N-terminus density (maps (**b**,**d**)) or lack the linker density (**c**). (4) The density for the extracellular loop (ECL) between TM helices 1 and 2 is weaker for maps (**c**,**e**). Lower panel: slices through the transmembrane regions of the maps (as indicated by the red dashed lines in the upper panel). Real-space refined atomic models that were generated are also displayed using ribbon representation in contrasting colors. The diameter of the micelles in maps (**a**,**c**) are larger; map (**e**) has the smallest micelle with some distortion from circular. Map (**a**) contains a central density (blue mesh, arrow) not accounted for by the real-space refined model. Other maps have only small features in the aqueous central cavity. Map (**e**) shows the most compact transmembrane organization.

**Figure 2 membranes-11-00923-f002:**
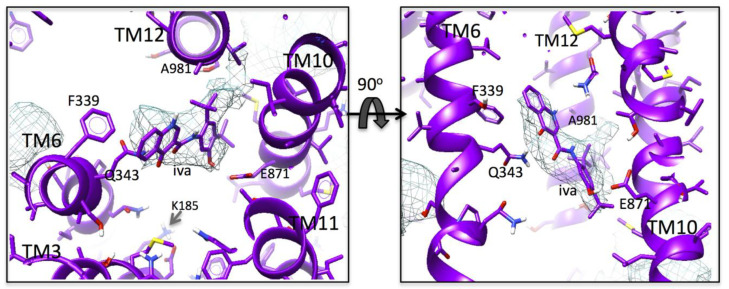
Additional density: Magnified, orthogonal views of additional density not accounted for by the fitted model in map *a* (blue mesh). The drug (iva) is modeled into the additional central density. The density is surrounded by residues in TM helices 6, 10, and 12 in the real-space refined atomic model. The 2,4 *tert*-butyl groups at one end of the ivacaftor molecule favored its orientation as shown within the rocket-shaped density. In this model, F339 and Q343 within TM6 could form π-π and H-bonding interactions with the ivacaftor quinoline group. Small (glycine) residues at 868 and 985 in TM helices 10 and 12 may be important for accommodating the bulky *tert*-butyl groups at the other end of the drug, and A981 may also satisfy hydrophobicity requirements for these groups. Similarly, H-bonding with E871 by the phenolic hydroxyl group of ivacaftor may occur in this fitting. K185 in TM3 extends up into the central cavity to within 10 Å of this group.

**Figure 3 membranes-11-00923-f003:**
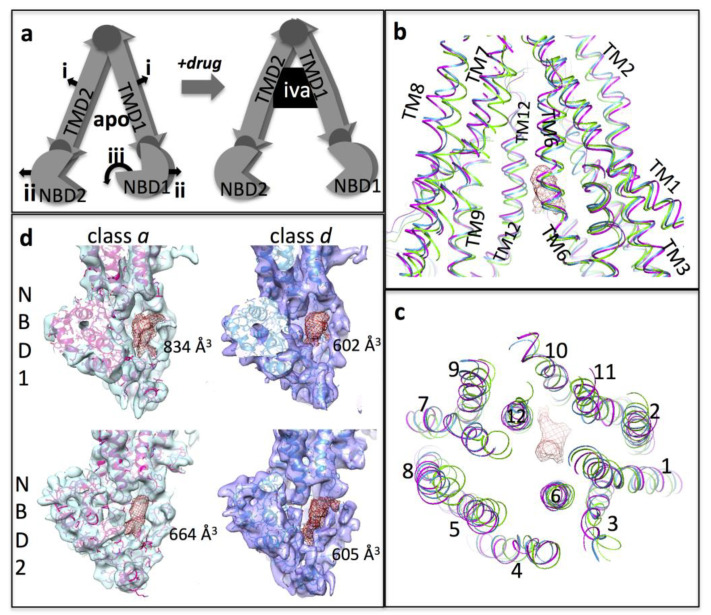
Allostery: (**a**) Cartoon representation of conformational changes discussed with regard to the presence of ivacaftor with P-gp. Left shows the assumed apo-state and indicates the movements of the TMDs and NBDs associated with the drug-bound state (right): (i) the TMD helices bow outwards and the central cavity enlarges. (ii) The NBDs become more separated. (iii) NBD1 rotates downwards. The small darker circles indicate hinge points between domains. (**b**,**c**) outward bulge of the TMDs in the atomic model from map a (pink ribbon) vs. models from map e (green) and map d (blue). The additional density associated with ivacaftor in map a is shown (red mesh). Panel b is a section viewed along the membrane plane; panel c is a section viewed from the extracellular surface. (**d**) NBD1 and NBD2 in experimental maps a and d viewed from between the NBDs (blue and purple semi-transparent surfaces). The real-space refined atomic models for P-gp fitted to the maps are in red and blue, respectively. The nucleotide-binding cavity is indicated with the red mesh and its volume is given in cubic Ångstrom.

**Figure 4 membranes-11-00923-f004:**
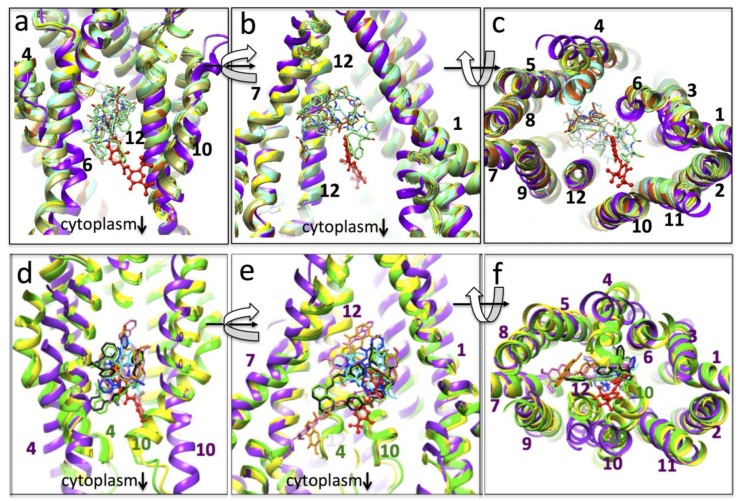
Location of drugs and inhibitors within the P-gp central binding cavity: (**a**–**c**): Murine P-gp models. Purple ribbon is the atomic model for map *a* in this paper (ivacaftor- red): yellow, PDBID 4q9h- apo-state; cyan, 4q9j- with QZ-Val; orange, 4q9i-QZ-Ala; taupe, 4q9k-QZ-Leu; green, 4q9l-QZ-Phe; turquoise, 4xwk-BDE-100. The modeled position of ivacaftor sits further towards the cytoplasm than the various inhibitors shown. Trans-membrane helices are numbered according to their position in the P-gp sequence. (**a**,**b**) are orthogonal views along the membrane plane; (**c**) is the view from the cytoplasm. Curved arrows indicate the 90° rotations involved. (**d**–**f**): Human and human/mouse chimera P-gp: no drug (6fn4, yellow ribbon); with zosuquidar (6fn1, green ribbon, green drugs, and black drugs-7a6f); tariquidar (7a6e, pink drugs); vincristine (7a69, cyan drug); taxol (6qex, blue drug); elacridar (7a6c, orange drugs). Wider separation of the murine NBDs necessitated alignment of the N- and C-terminal arms of the model (this paper, purple ribbon, red drug) with the human P-gp structures. Where major differences exist between the paths of transmembrane helices in human and murine models, they are numbered in both purple and green font.

**Table 1 membranes-11-00923-t001:** Summary of all the data for reciprocal space resolution assessment of the five maps as well as the real-space correlation coefficient (CC) between the atomic model and the relevant map.

Parameter.	Map *a*	Map *b*	Map *c*	Map *d*	Map *e*
Global Resolution (Å, FSC = 0.143) from unmasked half maps (CisTEM)	5.4	4.3	4.3	4.2	4.2
Global Resolution (Å, FSC = 0.143) from map to model fit (Phenix)	4.6	4.4	4.5	4.4	4.4
Resolution (Å) from Resmap with masked half maps (Mean, Mode)	5.0, 4.0	4.5, 4.0	4.6, 4.0	4.5, 4.0	4.4, 4.0
CC map to model fit (mean from 1182 residues, Phenix)	0.73	0.67	0.63	0.69	0.65

## Data Availability

Experimental density maps (*a*), (*e*), and the atomic models can be downloaded via the electron microscopy database (EMDB) and via the Protein Data Bank (RCSB) under the codes EMD-13059, 7OTG (map *a*), and EMD-13060, 7OTI (map *e*).

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
