# Peer review of "Structure of ABCB1/P-Glycoprotein in the Presence of the CFTR Potentiator Ivacaftor"

_membranes, 2021, doi:10.3390/membranes11120923_

Round 1

Reviewer 1 Report

This study conducted by Barbieri et al. exhibiting an interesting and through investigation on the P-gp conformation change in the presence of ivacaftor, a known competitive inhibitor. Detailed zoom in 2D interactive images were provided to reveal the bonding between P-gp and ivacaftor. Furthermore, the study on the NBD1 also explained the increased affinity for ATP binding when there were substrates present. Overall, this is excellent research, and would be more complete if the following suggestions and errors could be included and corrected in the revised version.

  1. Basically, P-gp protein is encoded by ABCB1 gene. So, for the description in the whole manuscript, the use of “P-gp” is suggested for representing the protein, while the use of “ABCB1” is for the gene.
  2. Several errors should be edited and the whole manuscript should be examined carefully, only a few was listed here for example:
    • In line 28, “ABCB1 (also named P-glycoprotein) is an ATP-dependent multi-drug transporter iof the ATP-binding cassette (ABC) family”, the iof is obviously an error.
    • In line 98, “were prepared as described in (4). Microsomal membranes were diluted to 2.5 mg/ml”, the (4) here is a reference? The style is different from other reference.
    • In line 228, ”summarised in Table I. Each 3D map could be generated from fewer particles than in”, I think here what the author mean is table 1, not table I.
  3. In line 331, “Similar changes, but with a smaller magnitude were also observed for NBD2 in the 331 atomic models.”, please also provide the observed data about NBD2 in this manuscript.
  4. In line 323-325, “When we compared map a’s model to that of map e (which was the most different – rmsd 1.40Å over 519 atom pairs within 2Å, 2.64Å over all 1182 atom pairs) the outwards bulging of the TM helices upon ivacaftor accommodation was more noticeable.”, the data should be included in the manuscript as the comparison to map d (Figure 3).

Author Response

REVIEWER 1

This study conducted by Barbieri et al. exhibiting an interesting and through investigation on the P-gp conformation change in the presence of ivacaftor, a known competitive inhibitor. Detailed zoom in 2D interactive images were provided to reveal the bonding between P-gp and ivacaftor. Furthermore, the study on the NBD1 also explained the increased affinity for ATP binding when there were substrates present. Overall, this is excellent research, and would be more complete if the following suggestions and errors could be included and corrected in the revised version.

We thank the reviewer for the positive and encouraging comments.

  1. Basically, P-gp protein is encoded by ABCB1 gene. So, for the description in the whole manuscript, the use of “P-gp” is suggested for representing the protein, while the use of “ABCB1” is for the gene.

We have changed ABCB1 to P-glycoprotein (abbrv= P-gp) throughout as suggested by the reviewer.

  1. Several errors should be edited and the whole manuscript should be examined carefully, only a few was listed here for example:
    • In line 28, “ABCB1 (also named P-glycoprotein) is an ATP-dependent multi-drug transporter iof the ATP-binding cassette (ABC) family”, the iof is obviously an error.
    • In line 98, “were prepared as described in (4). Microsomal membranes were diluted to 2.5 mg/ml”, the (4) here is a reference? The style is different from other reference.
    • In line 228, ”summarised in Table I. Each 3D map could be generated from fewer particles than in”, I think here what the author mean is table 1, not table I.

We thank the reviewer for picking up these errors and we have checked the manuscript again.

  1. In line 331, “Similar changes, but with a smaller magnitude were also observed for NBD2 in the 331 atomic models.”, please also provide the observed data about NBD2 in this manuscript.

Figure 3 has now been modified to show this. We also estimated the volume of the nucleotide binding cavity in both NBDs and report these measurements in the modified Figure 3.

  1. In line 323-325, “When we compared map a’s model to that of map e (which was the most different – rmsd 1.40Å over 519 atom pairs within 2Å, 2.64Å over all 1182 atom pairs) the outwards bulging of the TM helices upon ivacaftor accommodation was more noticeable.”, the data should be included in the manuscript as the comparison to map d (Figure 3).

Figure 3 has been modified to show the outward bulge of the TM helices. Referee 2 also suggested this as an amendment of the manuscript.

Reviewer 2 Report

The manuscript by Barbieri et al. reports on a structural study of ivacraftor binding to mouse ABCB1. The manuscript is well written, clear and easy to follow. Much effort is spend on the quality control of the data. The presented structures are of a 4-5 Angstrom resolution and are compared to previously solved structure of ABCB1. 

The N-terminal and the C-terminal half transporter are quite similar in their overall shape. Alignment of the single particles could therefore be a challenge. I might have missed it, but I could not find a description, how this issue was approached.

The extracellular loop 1 (EL1) of ABCB1 is much longer then the other ELs. I am wondering, if EL1 was resolved. It remains unclear from the manuscript and the SI. 

Inserting ivacraftor into the extended rocket-shaped density could in principle be done using two different orientations. A discussion, how one of the putative modes could be excluded, would be welcome. 

Further analysis of the change in the overall shape of ABCB1 conformation, supposedly induced by ivacraftor binding, would support one of the key finding. Figure 3a is of too low resolution for reporting on/supporting this finding. It might be useful to create a structural overlay of the 5 conformations, maybe by fitting on only one of the half transporters, if needed, to visualize the changed. 

Comparison to human ABCB1: In the beginning the authors highlight the importance of the low drug concentration used in this study. I am wondering, if the collapsed structure of the human ABCB1 structures might be a consequence of (much?) higher ligand concentrations. Would the authors be able to comment or have data that indicates a role for ligands in inducing such structural changes in response to increasing concentrations.

Minor points. 
Introduction, first line: "iof" should read "of".

Material an methods section. Several units are attached to the respective numbers. Please insert spaces.

Fourth last line of the discussion: "c lose" should read "close"

Author Response

The manuscript by Barbieri et al. reports on a structural study of ivacraftor binding to mouse ABCB1. The manuscript is well written, clear and easy to follow. Much effort is spend on the quality control of the data. The presented structures are of a 4-5 Angstrom resolution and are compared to previously solved structure of ABCB1. 

We thank the reviewer for the encouragement.

The N-terminal and the C-terminal half transporter are quite similar in their overall shape. Alignment of the single particles could therefore be a challenge. I might have missed it, but I could not find a description, how this issue was approached.

This is a good point. We have added some text to address this and have added Supplementary Figure 9 which illustrates how the asymmetric parts of the protein are apparent in projections and aid in discrimination between 0° and 180° rotated versions of the particles.

The extracellular loop 1 (EL1) of ABCB1 is much longer then the other ELs. I am wondering, if EL1 was resolved. It remains unclear from the manuscript and the SI. 

Yes, most of EL1 is resolved in map a, although the map to model correlation is worse for a few residues in this region (Supplementary Fig. 8).

Inserting ivacraftor into the extended rocket-shaped density could in principle be done using two different orientations. A discussion, how one of the putative modes could be excluded, would be welcome. 

We included some text description of using the fins of the rocket as possible correlating with the tert-butyl groups of the drug and have added some further text in the methods and results sections to cover this better. We have added some further quantitation using the correlation coefficients of the two alternative orientations vs the experimental density. The two alternative orientations have correlation coefficients (simulated map vs experimental map) of 0.938 (for one orientation) vs 0.960 (for the orientation with tert-butyl groups fitted to the fins of the rocket).

Further analysis of the change in the overall shape of ABCB1 conformation, supposedly induced by ivacraftor binding, would support one of the key finding. Figure 3a is of too low resolution for reporting on/supporting this finding. It might be useful to create a structural overlay of the 5 conformations, maybe by fitting on only one of the half transporters, if needed, to visualize the changed. 

We have taken Reviewer 2’s nice suggestion and modified Fig. 3 accordingly. Superimposition of all 5 models was too messy, but we superimposed 3 of the models (for maps a, d and e) to show the outward bulge of some of the TM helices.

Comparison to human ABCB1: In the beginning the authors highlight the importance of the low drug concentration used in this study. I am wondering, if the collapsed structure of the human ABCB1 structures might be a consequence of (much?) higher ligand concentrations. Would the authors be able to comment or have data that indicates a role for ligands in inducing such structural changes in response to increasing concentrations.

Yes, this is an excellent suggestion and we have added a few sentences to the end of the Discussion.

Minor points. 

Introduction, first line: "iof" should read "of".

Material an methods section. Several units are attached to the respective numbers. Please insert spaces.

Fourth last line of the discussion: "c lose" should read "close"

We thank the reviewer for highlighting these typos.

Reviewer 3 Report

In this study, Barbieri et al. used cryoEM to localize the binding site of ivacaftor on P-glycoprotein and analyze the conformational changes induced by drug binding. The authors purposely used sub-stoichiometric concentrations of ivacaftor in their samples, hoping that drug binding would cause a conformational change in ABCB1, and that they would be able to differentiate conformational populations (apo and drug-bound states) using 3D classification of the particles. To increase their chances to distinguish the different conformations, they also employed the Volta phase plate to collect high signal:noise data with minimal under focus. The methodological approaches and the findings are interesting but I have several concerns regarding the results and their interpretation.

Major:

  • Although it is stated in the abstract that ivacaftor increases the affinity of ABCB1 for ATP, I did not see any biochemical evidence that support this statement in the article….
  • The experiments in Fig. S1 (panels A and B) are not sufficiently explained and were unclear to me. Also, I doubt you could extrapolate a valid Kd from panel A, where no saturation is visible. Regarding panel B, is it possible that light scattering changes are due to a fraction of protein aggregation?
  • The authors localized the ivacaftor binding site based on an additional density visible in map a. Given the high affinity of ivacaftor (0.2 to 1 uM) for ABCB1, the authors should mutate one of the amino acids interacting with the drug (Q343 seems like a good candidate) and check if the mutation decreases the affinity. Such experiment would greatly strengthen and support the conclusions of the study
  • The authors suggest that ivacaftor binding induces and stabilizes conformational changes in ABCB1, including a wider separation of its two halves. I feel it is a bit counterintuitive because ivacaftor greatly stimulates the ATPase activity of ABCB1 and ATP hydrolysis occurs when the NBDs close. Could the authors comment on that? On the other hand, ivacaftor binding seems to facilitate a proper orientation of NBD1 relative to NBD2 (Fig. 3A), did the other transported substrates (Fig. 4) induced similar conformational effects?

Minor:

  • Page 4, section 3.1. Although references are given, it would help the readers to be more precise. Ivacafor was observed to be a competitive inhibitor, for which transported drug(s)? Also, I assume that the fluorescent dye was Hoechst 33342?
  • Page 4, section 3.1. “The Kd was estimated at 0.3 uM” From the reference quoted, I think you should refer to IC50 instead of Kd?

Author Response

In this study, Barbieri et al. used cryoEM to localize the binding site of ivacaftor on P-glycoprotein and analyze the conformational changes induced by drug binding. The authors purposely used sub-stoichiometric concentrations of ivacaftor in their samples, hoping that drug binding would cause a conformational change in ABCB1, and that they would be able to differentiate conformational populations (apo and drug-bound states) using 3D classification of the particles. To increase their chances to distinguish the different conformations, they also employed the Volta phase plate to collect high signal:noise data with minimal under focus. The methodological approaches and the findings are interesting but I have several concerns regarding the results and their interpretation.

Major:

  • Although it is stated in the abstract that ivacaftor increases the affinity of ABCB1 for ATP, I did not see any biochemical evidence that support this statement in the article….

Apologies for the sloppy wording here and we thank the reviewer for picking up this blunder on our part. We were aiming to link the structural work here to earlier work establishing that many ABC transporters, including P-gp, have a higher affinity for ATP and higher ATPase activity upon allocrite binding. As ATP concentrations in the cell are much higher than Km for ATPase activity, affinity will be much less relevant than Vmax. We have changed the abstract last sentence to: Conformational changes to the nucleotide binding domains are also observed, and may help to explain the stimulation of ATPase activity that occurs when transported substrate is bound in many ATP binding cassette transporters.

  • The experiments in Fig. S1 (panels A and B) are not sufficiently explained and were unclear to me. Also, I doubt you could extrapolate a valid Kd from panel A, where no saturation is visible. Regarding panel B, is it possible that light scattering changes are due to a fraction of protein aggregation?

We acknowledge that these data were explained briefly and have added some extra explanations and a reference where the CPM methodology is outlined. We also show the aggregated data for thermal unfolding at all the ivacaftor concentrations used in a modified Supplementary Fig. 1. The aim of these biochemical experiments was to show that ivacaftor interaction with murine P-gp was not much different to in human P-gp (where its biochemical parameters were explored in greater detail in the two published papers cited). This seems to be the case. The light scattering changes show a decrease at higher temperatures for P-gp, which we think is due to unfolding of the protein into the detergent micelles. Membrane protein aggregation is occasionally observed even with detergent present after long periods at >70°C but this is accompanied by a very strong increase in SLS at 266nm that is not observed in these experiments.

  • The authors localized the ivacaftor binding site based on an additional density visible in map a. Given the high affinity of ivacaftor (0.2 to 1 uM) for ABCB1, the authors should mutate one of the amino acids interacting with the drug (Q343 seems like a good candidate) and check if the mutation decreases the affinity. Such experiment would greatly strengthen and support the conclusions of the study

We agree that for full completion of these studies, mutagenesis of Q343 and one or two other residues in the vicinity of the proposed ivacaftor site will need to be carried out. However we would plead that these studies merit publication at this initial stage. In comparable recent structural studies of P-gp in the presence of drug from the Chang and Locher labs, such mutagenesis studies were not carried out yet. The corresponding mutation Q347A has been generated recently in human P-gp. This shows loss of transport function for the P-gp substrate CalceinAM but, interestingly, not for a fluorescent version of taxol. (J Mol Sci 2021 Aug 9;22(16):8561). Taxol binds in a slightly different position to ivacaftor (our Figure 4), but whether this will be the same site for the fluorescently-labelled version of taxol is unknown.

  • The authors suggest that ivacaftor binding induces and stabilizes conformational changes in ABCB1, including a wider separation of its two halves. I feel it is a bit counterintuitive because ivacaftor greatly stimulates the ATPase activity of ABCB1 and ATP hydrolysis occurs when the NBDs close. Could the authors comment on that?

We have added some discussion of this and thank the reviewer for prompting this.

  • On the other hand, ivacaftor binding seems to facilitate a proper orientation of NBD1 relative to NBD2 (Fig. 3A), did the other transported substrates (Fig. 4) induced similar conformational effects?

For the human P-gp structures with inhibitors and substrates bound, there is a minor opening up of NBD1 versus the apo state, but overall the structures for human P-gp as shown in Figure 4d-f are remarkably similar (but different at the level of individual amino acid side-chain rotamers). This similarity, even in the degree of opening of the two halves of the protein, may be due to conformational stabilisation by the antibody used in these studies.

For the murine P-gp structures shown in Fig. 4 a-c, with the exception of the model in the presence of ivacaftor, all the other NBD1 binding sites hardly change at all even at the level of amino acid side chain rotamers (which is very surprising). The main difference between these structures is the degree of opening of the inward-facing state. It is possible that the lack of NBD1 changes may be because these murine structures (with the exception of ivacaftor) are all for inhibited for P-gp, and may represent a locked conformation. In summary, as we discuss in the Introduction, we think the conformational changes we are seeing in the global P-gp conformation and in NBD1 in particular (with regard to the nucleotide binding site) are novel findings for this protein and will be interesting to the Membranes readership.

Minor:

  • Page 4, section 3.1. Although references are given, it would help the readers to be more precise. Ivacafor was observed to be a competitive inhibitor, for which transported drug(s)? Also, I assume that the fluorescent dye was Hoechst 33342?

This is correct, these changes have been made. Hoechst 33342 in one study, digoxin in the other. Many thanks.

  • Page 4, section 3.1. “The Kd was estimated at 0.3 uM” From the reference quoted, I think you should refer to IC50 instead of Kd?

Correct. Many thanks for picking this up.

Round 2

Reviewer 2 Report

The authors have successfully addressed all my comments and concerns.

Reviewer 3 Report

Except for the site-directed mutagenesis experiment that they chose to not perform, the authors responded to most of the other comments.